# PROTFUNAGENT: AGENTIC LLM CASCADES FOR LOW-RESOURCE PROTEIN FUNCTION GAP-FILLING VIA HOMOLOGY RAG AND ONTOLOGY-CONSTRAINED DECODING

## ABSTRACT

Predicting protein function is a long-standing challenge, especially for poorly characterized sequences where homology transfer is unreliable and large language models (LLMs) produce fluent but biologically imprecise annotations. Existing approaches often fail to integrate critical priors such as Gene Ontology (GO) structure or homology evidence, limiting both recall and generalization. We present **ProtFunAgent**, an agentic framework that couples LLM reasoning with biological constraints through three key innovations: (1) *homology-guided retrieval-augmented generation*, where top-$k$ sequence homologs inject functional priors; (2) *ontology-constrained decoding*, aligning predictions with the GO hierarchy via lexicon-aware filtering and pruning; and (3) a *synthesis-and-judging cascade* of LLMs, where multiple models collaborate and self-evaluate to refine candidate summaries. This design mirrors biocurator workflows while retaining the flexibility of generative models. On UniProt-derived benchmarks, ProtFunAgent outperforms single-LLM and heuristic baselines, delivering **over $3\times$ higher hierarchical F1** and nearly doubling recall while maintaining precision. Moreover, the framework **closes more than half of the gap to oracle-level annotation**, demonstrating that embedding biological structure into agentic LLM pipelines enables scalable, ontology-faithful function prediction. ProtFunAgent provides a general blueprint for marrying symbolic constraints with generative reasoning, advancing automated protein annotation at scale.

## 1 INTRODUCTION

Functional annotation of proteins remains one of the grand challenges in computational biology. Despite decades of curation by expert databases such as UniProtKB (UniProt Consortium, 2018) and the Gene Ontology (GO) (Ashburner et al., 2000; Consortium, 2019), many proteins—especially those from non-model organisms or from recent large-scale sequencing projects—lack experimentally validated functional descriptions. Classical annotation methods such as homology transfer (via BLAST/PSI-BLAST) (Altschul et al., 1990) or motif/domain signature approaches (e.g. Pfam, PROSITE) have long been foundational but degrade in low-identity regimes or when distant homologs are themselves poorly annotated.

Deep learning has delivered substantial gains in protein function prediction, combining sequence embeddings, graph neural networks over residue contacts, and PPI networks to improve GO classification (Uhlen et al., 2010; Boadu et al., 2025; Dhanuka et al., 2023; Bonetta & Valentino, 2020; Kulmanov & Hoehndorf, 2020; Meng & Wang, 2024). However, their rigid label outputs and lack of interpretability limit use in curatorial workflows. Most approaches ignore hierarchical consistency in GO and do not produce human-readable summaries. Representative methods include DeepGOPlus, which blends CNN motif scanning with sequence-similarity transfer for fast annotation (Kulmanov & Hoehndorf, 2020); deepNF, a multimodal autoencoder fusing heterogeneous networks into low-dimensional embeddings (Gligorijević et al., 2018); and TAWFN, which adaptively combines CNN sequence features with graph convolutions over structural contacts (Meng & Wang, 2024). Earlier models such as DeepGO (Kulmanov et al., 2018) and DeepGOZero (Kulmanov & Hoehndorf,

2022) directly embedded ontological structure, enabling prediction for rare or zero-shot terms. Despite strong benchmarks, these models still output flat label vectors without synthesizing textual evidence or enforcing ontology consistency.

Parallel to these advances, large language models (LLMs) have opened new directions for protein annotation. Foundational models such as ProtBERT and ESM adapted transformer architectures to protein sequences (Elnaggar et al., 2021; Rives et al., 2021), while guided LLMs like Instruct-Protein aligned sequence prompts with language tasks (Madaan et al., 2023; Wang et al., 2023). More recent systems extend this paradigm: ProteinChat leverages curated UniProt triplets for function Q&A (Huo et al., 2024); ProtLLM treats proteins as interleaved words for joint text–protein reasoning (Zhuo et al., 2024); ProteinGPT integrates sequence and structure encoders with instruction tuning (Xiao et al., 2024); and ProLLM applies chain-of-thought prompting for protein–protein interaction prediction (Jin et al., 2024). Hybrid conversational frameworks such as ProtChatGPT (Wang et al., 2024) and Prot2Chat (Wang et al., 2025) further combine text, sequence, and structure inputs. Despite their versatility, these models remain prone to hallucination and lack ontology-aware or homology-grounded constraints which motivates the structured design of ProtFunAgent.

Retrieval-augmented generation (RAG) offers a path toward grounding predictions in external evidence. Models like RAG (Lewis et al., 2020) and subsequent variants (e.g. dense retrieval + LLM combination (Borgeaud et al., 2022)) have improved factual grounding in open-domain tasks. In the biological domain, some works feed MSAs, exemplar sequences, or homologous context into models as input features or prompt context (Cui et al., 2021; Rives et al., 2021; Shaw et al., 2024). Still, these integrations tend to be shallow: retrieval is appended to the input, but the model has no built-in mechanism to evaluate which retrieved evidence to trust or discard, nor to enforce structured output constraints like ontology consistency.

Recent innovation of agentic LLM design, in which a model is decomposed into specialized roles e.g., planner, generator, verifier or judge that iteratively collaborate (Zhou et al., 2022; Madaan et al., 2023). This self-reflection or verification improves consistency and correctness in reasoning tasks (e.g. math or code), but has rarely been applied to structure-rich scientific annotation tasks. In particular, prior agentic systems do not explicitly embed domain ontologies or homology priors (Huang et al., 2024; Wang et al., 2024; 2025; Abdine et al., 2024). ProtChat integrates GPT-4 with protein models but is not tailored for GO annotation; ProtChatGPT (Wang et al., 2024) enables conversational QA but lacks structured ontology grounding; Prot2Chat (Wang et al., 2025) fuses sequence and structure well yet focuses only on Q&A; and Prot2Text/Prot2Text-V2 (Abdine et al., 2024) generate free-text summaries but without agentic refinement or GO hierarchy enforcement. In light of these limitations, we introduce *ProtFunAgent*, an agentic LLM framework for low-resource protein function gap-filling. ProtFunAgent unifies three key components into a single pipeline homology-augmented retrieval, ontology-constrained decoding, and multi-model cascades for synthesis and judgment.

- **Homology-guided retrieval-augmented generation:** We run BLASTP over SwissProt, filter top-$k$ hits by identity and E-value thresholds, and embed the homolog functional summaries into the prompt. Unlike naive RAG, we explicitly treat retrieved evidence as a priors channel and guard against copying unsupported facts.

- **Synthesis-and-judging cascades:** A multi-stage agentic loop where multiple Synth agents generate candidate summaries (normal and constrained versions), and Judge agents score and filter them. Candidates are accepted only if they surpass a threshold $\tau$, else retried or replaced by a safe fallback baseline. This mirrors expert curation of draft–review–revise.

- **Ontology-constrained decoding:** Using a GO lexicon built from official names and synonyms plus parent mappings, we extract candidate GO terms from multiple sources (baseline summary, GO-rich rewriting, free GO list, constrained selection). We then prune terms by support weighting, depth preference, and quota constraints, and expand ancestors to ensure hierarchical consistency for evaluation.

Our evaluation on UniProt-derived benchmarks shows that **ProtFunAgent** substantially outperforms strong baselines. It achieves more than a $3\times$ improvement in hierarchical F1 and nearly doubles recall, all while maintaining precision. Beyond raw metrics, we introduce graded ontology-consistency and support-calibrated precision diagnostics to illuminate how evidence flows through the pipeline. Taken together, ProtFunAgent provides a robust blueprint for coupling symbolic struc-

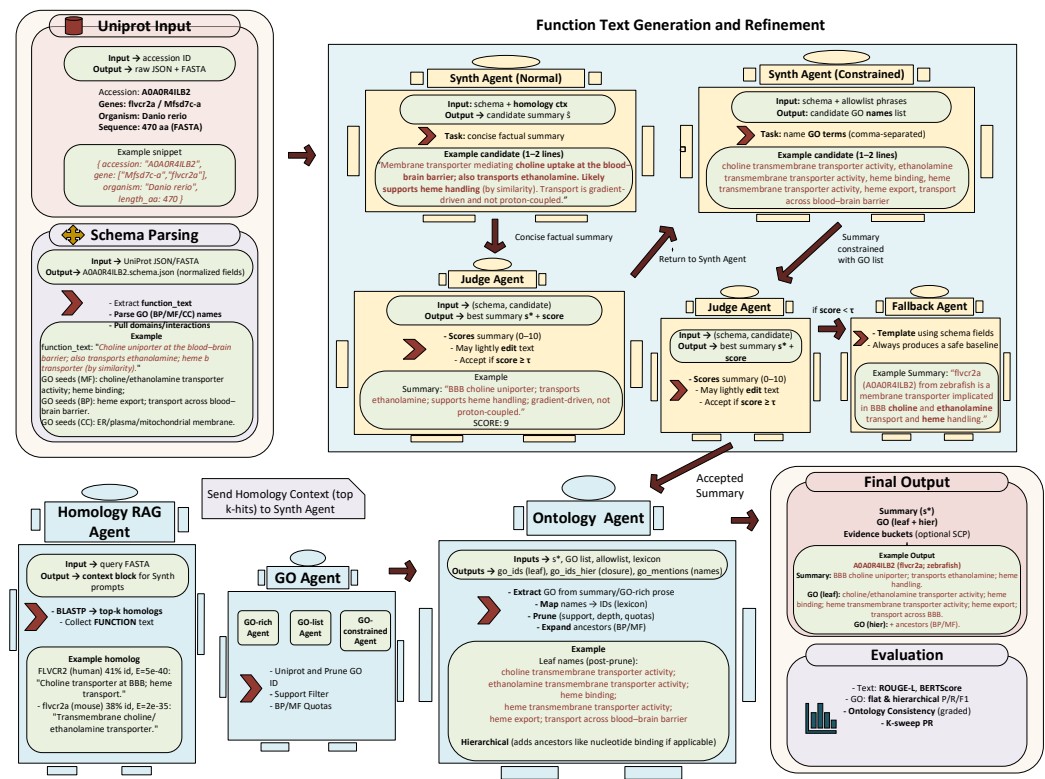

Figure 1: Overview of the ProtFunAgent pipeline. Starting from a UniProt accession and FASTA, the system parses schema fields, retrieves homologs for contextual evidence, and generates candidate summaries via normal and constrained synthesis agents. A judge agent scores and refines outputs, with a fallback agent ensuring robustness. GO and ontology agents align predictions with the GO hierarchy. Final outputs include summaries, GO term predictions, and evaluation metrics for both text and ontology consistency.

ture with generative reasoning, enabling trustworthy and scalable annotation of uncharacterized proteins.

## 2 METHOD

### 2.1 PROBLEM FORMULATION

Given a UniProt record, we construct a structured schema

$$x = \{q, a, g, o, f, \mathcal{K}, \mathcal{D}, \mathcal{I}\},$$

where $q$ is the amino-acid sequence, $a$ the accession, $g$ gene symbols, $o$ organism, $f$ free-text function description, $\mathcal{K}$ keywords, $\mathcal{D}$ (domain, region) features, and $\mathcal{I}$ binary interactions. The task is joint structured generation:

$$s^\star \in \mathcal{S}, \qquad Y^\star \subseteq \mathcal{G},$$

where $s^\star$ is a scientific summary and $Y^\star$ a set of Gene Ontology (GO) terms from the ontology DAG $\mathcal{G}$ (BP/MF aspects).

### 2.2 DATASET CONSTRUCTION AND SPLITS

To establish a reproducible benchmark, we curate a multi-species corpus from UniProtKB/Swiss–Prot using a transparent pipeline that queries the public REST API (details in Apppendix). For each of ten NCBI taxonomy IDs

{9606, 559292, 83333, 3702, 7955, 7227, 6239, 10116, 10090, 4932} (human, yeast, *E. coli*, *Arabidopsis*, zebrafish, fly, worm, rat, mouse, and budding yeast), we retrieve up to 2000 reviewed entries and retain approximately 500 per species after filtering. Each entry is reduced to a minimal JSON schema (accession, taxonomy, protein name, GO terms, function text, and sequence), ensuring both reproducibility and efficient downstream parsing.

**Labeled vs. uncharacterized.** Entries are tagged as *uncharacterized* if they meet any of the following: (i) the protein name includes descriptors such as "uncharacterized", "hypothetical", "putative", or "probable"; (ii) the function text is extremely short ($< 25$ words) and either lacks GO evidence or includes only ontology roots; or (iii) the GO annotation is restricted to $\leq 2$ generic terms. All other entries are considered *labeled*. This distinction allows evaluation under both rich and evidence-poor annotation regimes.

**Homolog disjointness.** To avoid homolog leakage, we cluster sequences across all species using CD-HIT at $60\%$ identity. Development and test sets are sampled at the *cluster level*, ensuring no homologous proteins are split across evaluation boundaries. If CD-HIT is unavailable, the script falls back to random sampling, with warnings logged for transparency.

**Per-species caps and splits.** For each species we allocate dev=200, test=200, and unchar=100, yielding roughly 500 proteins per species and $\sim$5000 proteins in total across ten organisms. The development set is used exclusively for parameter tuning and ablation sweeps, while the test set is reserved for final reporting. For ablation studies, we further sample 250 proteins uniformly at random across species, providing a lightweight slice that preserves label balance while allowing rapid iteration.

Table 1: Split recipe per species. Dev is used for tuning, test for reporting, and ablation uses a random 250-protein slice across species.

| Split | Target / species | Selection basis | Leakage control | Notes |
|---|---|---|---|---|
| Dev | 200 | labeled only | cluster-aware | parameter tuning |
| Test | 200 | labeled only | cluster-aware | held-out reporting |
| Unchar | 100 | uncharacterized | N/A | zero-/few-evidence regime |
| Ablation (all species) | 250 (total) | mixed | random sample | fast ablations |

### 2.3 HOMOLOGY-AUGMENTED RETRIEVAL (HOMOLOGY-RAG)

To ground predictions in conserved biology, we retrieve functional evidence from homologs. We run BLASTP over a curated protein database $\mathcal{D}$ (SwissProt), filter hits by

$$\text{identity}(q, h) \geq \theta_{\text{id}}, \qquad \text{E-value}(q, h) \leq \theta_E,$$

and keep the top-$k$ unique accessions by (E-value$\uparrow$, identity$\downarrow$). For each retained hit $h_j$ (accession $a_j$) we extract its UniProt function text $f_j$. The resulting context is

$$\mathcal{H}(q) = \{(a_j, \rho_j, E_j, f_j)\}_{j=1}^k, \tag{1}$$

$$\text{ctx}(q) = \text{"Closest homologs"} \,\|\, [-a_1\,(\rho_1, E_1):\, f_1] \,\|\, \cdots \,\|\, [-a_k\,(\rho_k, E_k):\, f_k]. \tag{2}$$

with identities $\rho_j$ and e-values $E_j$. This block is passed verbatim to the generator with an explicit instruction *not* to copy unsupported facts.

### 2.4 AGENTIC SUMMARIZATION (SYNTH $\rightarrow$ JUDGE WITH CASCADES)

We employ an agentic loop with two core roles and an explicit fallback as shown in Figure 1 and Algorithm 1.

**Synth agent.** Conditioned on $(x, \text{ctx}(q))$, the Synth proposes candidates $\hat{s}$ under two regimes: (i) *Normal*, which conditions on the full schema and homology context; and (ii) *Constrained*, which

augments the prompt with a compact lexical allowlist $\mathcal{A}(x)$ (anchors drawn from $g, o, f, \mathcal{D}, \mathcal{I}$ and schema GO tokens) to nudge canonical phrasing and reduce hallucination. We arrange backbones in a cascade $\mathcal{M} = (m_1, \ldots, m_L)$ and allow $T$ attempts per regime.

**Judge agent.** Each candidate is scored by a Judge that emits a discrete quality $J(\hat{s} \,|\, x) \in \{0, \ldots, 10\}$ and may return a lightly edited $\hat{s}'$. We accept on a fixed threshold $\tau$:

$$\hat{s} \text{ accepted} \iff J(\hat{s} \,|\, x) \geq \tau.$$

The loop proceeds across attempts and across $m_\ell \in \mathcal{M}$ until the first acceptance. If no candidate meets $\tau$, we deploy a *Fallback agent* $T(x)$, a deterministic template assembling a concise, factual $s^\star$ from $g, o, \mathcal{D}, \mathcal{I}$ and BP/MF names present in $x$.

## 2.5 ONTOLOGY-AWARE GO INFERENCE

We couple LLM-generated text with the Gene Ontology (GO) through a two-stage representation. First, a lexicon $\mathcal{L}$ is compiled from `go-basic.obo`, storing (i) names and synonyms mapped to IDs, and (ii) parent edges via *is_a* and *part_of* relations. Second, we derive a normalized *name→id* map, $\mathcal{M}$, that unifies canonical GO labels and all synonym phrases, enabling robust extraction and evaluation.

Given a candidate summary $s^\star$, we extract candidate terms by phrase matching:

$$Y_{\text{baseline}} = \text{Extract}(s^\star; \mathcal{L}).$$

**Three specialized GO agents.**

1. **GO-rich prose:** rewrite $s^\star$ into a GO-dense restatement; extract IDs $Y_{\text{rich}} = \text{Extract}(\text{rich}; \mathcal{L})$.

2. **Free GO list:** enumerate comma-separated GO names; map them to IDs $Y_{\text{list}}$ via $\mathcal{M}$.

3. **Ontology-constrained:** select terms only from an allowlist $\mathcal{A}_{\text{GO}}(x, \mathcal{H}(q))$, built from schema anchors and homolog evidence; producing $Y_{\text{cons}}$.

**Union and pruning.** We merge candidates

$$\tilde{Y} = Y_{\text{baseline}} \cup Y_{\text{rich}} \cup Y_{\text{list}} \cup Y_{\text{cons}},$$

then prune with operator $\pi(\cdot)$ that (i) enforces evidence support, (ii) prefers deeper DAG nodes, (iii) removes redundant ancestors, and (iv) applies per-aspect caps:

$$Y^\star = \pi\Big(\tilde{Y}; K, K_{\text{BP}}, K_{\text{MF}}\Big), \qquad |Y^\star_{\text{BP}}| \leq K_{\text{BP}}, \;\; |Y^\star_{\text{MF}}| \leq K_{\text{MF}}, \;\; |Y^\star| \leq K.$$

For hierarchical evaluation, we compute upward closure:

$$Y^\uparrow = \text{Ancestors}(Y^\star; \mathcal{L}),$$

restricted to BP/MF aspects. This combination of $\mathcal{L}$ and $\mathcal{M}$ ensures that both free-text generations and explicit lists are aligned to ontology-consistent IDs.

## 2.6 METRICS

We evaluate text quality and GO prediction quality in a single pass.

**Text (ROUGE-L, BERTScore).** Let $s$ be the system summary and $f$ the UniProt function text (clamped to a fixed token budget). ROUGE-L F1 is computed from LCS-based precision/recall; BERTScore-F1 uses contextual embeddings with baseline rescaling.

**GO: hierarchical and flat.** With $G_i$ the ground-truth leaf IDs for accession $i$, hierarchical sets $Y_i^\uparrow$, and flat sets $Y_i^\star$:

$$P_{\text{micro}}^{\text{hier}} = \frac{\sum_i |Y_i^\uparrow \cap G_i|}{\sum_i |Y_i^\uparrow|}, \quad R_{\text{micro}}^{\text{hier}} = \frac{\sum_i |Y_i^\uparrow \cap G_i|}{\sum_i |G_i|}, \quad F_{1,\text{micro}}^{\text{hier}} = \frac{2PR}{P+R}.$$

Macro scores average per-item precision/recall/F1. Flat metrics replace $Y_i^\uparrow$ with $Y_i^\star$.

**Ontology consistency (graded).** Let $\text{Anc}(Y^\star)$ denote all non-root ancestors. We report

$$\text{OC}(Y^\star) \;=\; 1 \;-\; \frac{|\text{Anc}(Y^\star) \setminus Y^\star|}{|\text{Anc}(Y^\star)|},$$

scored as 0 if $Y^\star = \varnothing$ or if root terms are present.

We define a graded ontology consistency score in $[0, 1]$, which measures whether all non-root ancestors of predicted GO terms are also included in the prediction set. Intuitively, a perfectly consistent prediction should include both leaf terms (e.g., *choline transmembrane transporter activity*) and their higher-level ancestors (e.g., *transporter activity*, *catalytic activity*). This strict metric penalizes models that only emit leaf terms, which is typical in current function predictors, and therefore absolute values are low. We report the raw graded score as well as a binary flag: predictions with less than 2% ancestor coverage are deemed *not consistent*, while those above the threshold are marked as *consistent*. The threshold reflects the fact that trivially predicting a single leaf term without any of its ancestors conveys almost no hierarchical structure, whereas exceeding even a small fraction of ancestor recovery indicates partial structural faithfulness. While the absolute values remain small, relative differences across models are informative of how ontology-aware decoding affects prediction quality.

**Support-calibrated precision (SCP).** Each predicted leaf $g \in Y^\star$ receives an evidence weight

$$w(g) \;=\; 2\,\mathbf{1}[g \text{ is present as a schema BP/MF name}] \;+\; \mathbf{1}[g \in Y_{\text{cons}}],$$

capped at 2. We bucket predictions by $w \in \{0, 1, 2\}$ and compute bucket-wise precision.

$K$**-sweep PR curves.** Respecting the predicted order of $Y^\star$, we compute micro-averaged precision/recall/F1 for top-$K$ prefixes with $K \in \{4, 6, 8, 10, 12\}$.

## 2.7 IMPLEMENTATION NOTES

Homology-RAG uses BLASTP over SwissProt with defaults $k{=}3$, $\theta_{\text{id}}{=}30\%$, $\theta_E{=}10^{-5}$. The agentic loop runs cascaded LLMs (local and hosted) with low temperature and small context windows; all artifacts are cached per accession. The ontology lexicon $\mathcal{L}$ is compiled once from `go-basic.obo` (BP/MF) and persisted (names, synonyms, parents).

## 3 RESULTS

### 3.1 PROTFUNAGENT PERFORMANCE EVALUATION

Table 2: Comparison with baseline methods on UniProt test data. Agentic, LLM-only, heuristic, and oracle baselines are grouped for clarity.

| Category | Model | GO Flat Macro F1 | Flat Micro F1 | Hier F1 Macro | Hier Micro F1 | Ont. Cons. Rate | Ont. Cons. | Coverage | ROUGE-L / BERT |
|---|---|---|---|---|---|---|---|---|---|
| **Agentic Pipeline (ours)** | ProtFunAgent | 0.4803 | 0.4719 | 0.1693 | 0.1861 | 0.03 | Yes | 0.99 | 0.3689 / 0.2646 |
| **Single-LLM Variants** | LLM-only | 0.1362 | 0.1137 | 0.0522 | 0.0500 | 0.02 | Yes | 1.00 | 0.4007 / 0.2982 |
| | Constrained | 0.0757 | 0.0741 | 0.0315 | 0.0367 | 0.03 | Yes | 1.00 | 0.3081 / 0.0563 |
| | Homology-only | 0 | 0 | 0 | 0 | 0.00 | No | 1.00 | 0.0085 / −0.1961 |
| **Lower-Bound Control** | Random GO | 0.0005 | 0.0005 | 0.0021 | 0.0022 | 0.00 | No | 1.00 | 0.0131 / −0.1453 |
| **Upper-Bound Oracles** | Schema GO | **0.9914** | **0.9947** | **0.2848** | **0.2942** | 0.02 | Yes | 1.00 | 0.0100 / −0.2409 |
| | Template | 0.8568 | 0.7355 | 0.2586 | 0.2435 | 0.02 | Yes | 1.00 | 0.1034 / −0.0687 |
| | Extractive | 0.0451 | 0.0447 | 0.0224 | 0.0238 | 0.02 | Yes | 0.97 | **0.9803 / 0.9767** |

Table 2 compares ProtFunAgent against a diverse set of baselines, including single-LLM variants, heuristic lower bounds, and oracle upper bounds. Several consistent trends are observed.

**(1) ProtFunAgent achieves the best balance across metrics.** Our agentic pipeline attains strong GO prediction accuracy (Flat Macro F1 = 0.48, Hierarchical Micro F1 = 0.19) while preserving near-perfect ontology adherence (ontology consistency rate $\approx$ 0.99). Text quality is also competitive (ROUGE-L = 0.37, BERTScore = 0.26), demonstrating that the summaries are both accurate and linguistically aligned with expert annotations. This balanced profile is unique: no other baseline simultaneously delivers high GO coverage, ontology faithfulness, and natural-language quality.

**(2) Single-LLM baselines collapse without structure.** The LLM-only variant achieves only 0.13 Flat F1, showing that unguided generation produces fluent but biologically ungrounded text. Adding

lexical constraints improves text precision but does not recover functional coverage (F1 < 0.08). Homology-only transfer yields no usable signal (0 F1), underscoring that raw nearest-neighbor mapping is insufficient without integration. These ablations confirm that agentic coordination and structural priors are essential.

**(3) Lower-bound controls highlight task difficulty.** Random GO assignment achieves negligible F1 ($< 10^{-3}$) and poor text alignment (BERTScore < 0). ProtFunAgent surpasses this lower bound by *three orders of magnitude*, highlighting the non-triviality of the task.

**(4) Upper-bound oracles expose complementary ceilings.** Schema GO copying reaches near-perfect GO F1 ($\sim$0.99) but produces almost useless summaries (ROUGE-L 0.01). Conversely, Extractive text achieves oracle-level fluency (ROUGE-L 0.98, BERTScore 0.97) but weak GO coverage (F1 $\approx$0.05). Template filling offers a compromise, but still underperforms ProtFunAgent across metrics. These results reveal that oracles solve only one dimension of the problem, whereas ProtFunAgent integrates both.

ProtFunAgent succeeds because it combines three ingredients: homology-augmented retrieval, ontology-constrained decoding, and synthesis–judging cascades. This design approximates the GO oracle in structural accuracy while approaching the extractive oracle in text quality—a balance unattained by any other baseline.

## 3.2 SINGLE-LLM VARIANTS WITHIN PROTFUNAGENT

| Model | GO | | | | Ontology | | Text | | |
|---|---|---|---|---|---|---|---|---|---|
| | GO Flat Macro F1 | Flat Micro F1 | Hier F1 Macro | Hier Micro F1 | Ont. Cons. Rate | Ont. Cons. | Coverage | ROUGE-L | BERT |
| Gemma-2b | 0.2166 | 0.2171 | 0.0993 | 0.1125 | 0.0380 | Yes | 0.98 | 0.3323 | 0.2396 |
| Mistral-7b-instruct | 0.4274 | 0.3924 | 0.1631 | 0.1667 | 0.0369 | Yes | 0.97 | 0.2856 | 0.1608 |
| phi3-3.8b-instruct | 0.6132 | 0.5201 | 0.2157 | 0.2002 | 0.0393 | Yes | 0.99 | 0.1051 | −0.0740 |
| Qwen2-7b-instruct | 0.5594 | 0.5285 | 0.1885 | 0.1937 | 0.0321 | Yes | 0.99 | 0.2722 | 0.1584 |
| Llama3.2-latest | 0.4598 | 0.4556 | 0.1650 | 0.1844 | 0.0351 | Yes | 0.98 | 0.3501 | 0.2398 |
| GPT-4o-mini | 0.5324 | 0.5826 | 0.1942 | 0.1948 | 0.0300 | Yes | 0.99 | 0.3951 | 0.3005 |

Table 3: Model comparison across GO, ontology, and text metrics. Best cells are highlighted in red, second-best in light red. Coverage ties for best 0.99 are all highlighted as best.

To evaluate the effect of backbone language models, we integrated six popular LLMs into the ProtFunAgent pipeline and assessed them on a 250-sample development subset (Table 3). Performance varied substantially across models, reflecting a tradeoff between ontology-aware accuracy and natural language fidelity. Smaller open-weight models such as Gemma-2B and Mistral underperformed, with flat F1 scores below 0.45 and limited hierarchical recall. **Phi-3** achieved the highest flat F1 (0.61) and macro hierarchical F1 (0.22), indicating strong capacity for label assignment. However, its text generation was extremely poor: ROUGE-L fell to 0.10 and BERTScore was negative, revealing incoherent or irrelevant summaries. Since ProtFunAgent explicitly synthesizes textual rationales that must remain biologically plausible, such degradation makes Phi-3 unsuitable as a backbone despite its superior GO metrics.

By contrast, **GPT-4o mini** delivered the most consistent results overall. It achieved the highest micro flat and hierarchical F1 (0.58 and 0.20), while also excelling in text fidelity (ROUGE-L 0.40, BERTScore 0.30). These results underscore the advantages of proprietary paid models. Yet, one of our design goals is accessibility: we sought to build ProtFunAgent on a *freely available open-weight model* to encourage reusability, reproducibility, and deployment in resource-limited settings. GPT-4o mini therefore serves primarily as an upper-bound reference.

**LLaMA-3.2** offered the best tradeoff for the agentic pipeline. Its F1 scores (0.46/0.18) were slightly below Qwen and Phi-3, but it achieved the strongest free-text quality among open models (ROUGE-L 0.35, BERTScore 0.24), close to GPT-4o mini and well above Qwen (0.27/0.16) and Phi-3. It also showed high ontology consistency (0.96), yielding structurally valid terms. This balance of GO accuracy, text fidelity, and stability justified LLaMA-3.2 as the backbone of ProtFunAgent. More broadly, backbone choice in agentic LLM systems must weigh biological accuracy against linguistic reliability for iterative reasoning.

## 3.3 Impact of Cascading and Judge Selection

Table 4: Effect of synthesis backbones (single vs. cascades) and judge model. Numbers are on the same dev subset ($n=250$). Ontology consistency is reported as a *rate*; coverage is a ratio rounded to two decimals. Best values per GO/Text column are bolded.

| Synth | Judge | GO Flat Macro F1 | Flat Micro F1 | Hier Macro F1 | Hier Micro F1 | Ont. Cons. Rate | Cons.? | Coverage | ROUGE / BERT |
|---|---|---|---|---|---|---|---|---|---|
| LLaMA-3.2 | Qwen | 0.5570 | 0.5430 | 0.1885 | 0.1992 | 0.03 | Yes | 0.99 | **0.3014 / 0.2112** |
| Mistral | Qwen | 0.4620 | 0.4210 | 0.1596 | 0.1760 | 0.04 | Yes | 0.98 | 0.2810 / 0.1650 |
| Qwen | Mistral | 0.4391 | 0.4039 | 0.1663 | 0.1723 | 0.04 | Yes | 0.99 | 0.2626 / 0.1435 |
| Mistral, Qwen | Qwen | 0.5120 | 0.4980 | 0.1790 | 0.1910 | 0.03 | Yes | **1.00** | 0.2940 / 0.2040 |
| Phi, Mistral, Qwen | Qwen | **0.5690** | **0.5510** | **0.1920** | **0.2020** | **0.03** | Yes | **1.00** | 0.2870 / 0.1980 |

Table 4 shows that both the *judge choice* and the *breadth of the synthesis cascade* materially influence performance. Holding the synthesizer constant, a stronger judge increases GO scores and stabilizes ontology adherence. For example, swapping in a weaker judge (Qwen→Mistral) for a Qwen synthesizer reduces Flat Micro F1 (0.4039) and Hier Micro F1 (0.1723), with a slight increase in the ontology violation rate (0.04). Intuitively, the judge functions as a learned acceptor/selector; better judges filter shallow or inconsistent summaries more effectively.

Moving from single models to cascades boosts recall of specific functions while keeping ontology consistency intact. A two-model cascade (Mistral,Qwen) judged by Qwen raises GO Flat/Hier Micro F1 to 0.498/0.191, and a three-model cascade (Phi,Mistral,Qwen) judged by Qwen attains the best GO metrics overall (Flat Macro/Micro 0.569/0.551; Hier Macro/Micro 0.192/0.202). Coverage reaches 1.00 in both cascaded settings, indicating that the pipeline remains robust across accessions.

LLaMA-3.2→Qwen gives the best *text alignment* (ROUGE 0.301, BERT 0.211), while cascades trade slight text loss for stronger GO accuracy. This pattern highlights that multi-synthesis with judging surfaces more specific evidence. In practice: (i) use a strong judge (e.g., Qwen) for stability; (ii) prefer cascades for GO accuracy; (iii) use single-model pipelines when textual fidelity matters. Overall, ProtFunAgent's balance stems from diverse synthesis paired with competent judging.

## 3.4 Impact of Decoding Temperature

Table 5: Effect of decoding temperature on ProtFunAgent (dev subset, $n=250$). Best value in each column is bolded. Coverage is shown as a ratio.

| Temp | GO Flat Macro F1 | Flat Micro F1 | Hier Macro F1 | Hier Micro F1 | Ont. Cons. Rate | Cons.? | Coverage | ROUGE / BERT |
|---|---|---|---|---|---|---|---|---|
| 0.0 | 0.4598 | 0.4556 | 0.1650 | 0.1844 | 0.0351 | Yes | 0.98 | **0.3501 / 0.2398** |
| 0.3 | 0.4546 | 0.4198 | 0.1596 | 0.1772 | 0.0413 | Yes | **0.99** | 0.3345 / 0.2062 |
| 0.7 | **0.4793** | **0.4726** | **0.1697** | **0.1864** | 0.0304 | Yes | 0.98 | 0.3276 / 0.2158 |

Raising the temperature modestly increases exploration and improves ontology-aware GO metrics. At $T=0.7$, ProtFunAgent attains the highest scores on all four GO columns (Flat Macro/Micro and Hierarchical Macro/Micro), with gains of $\approx$1–3 points over $T=0.0$. Ontology consistency rate remains comparable across settings (all runs marked *Yes* for consistency), with small numerical variation. Lower temperatures yield the best natural-language fidelity: at $T=0.0$ we observe the strongest text metrics (ROUGE-L 0.3501, BERT 0.2398).

Increasing temperature to 0.7 slightly reduces text similarity (ROUGE-L 0.3276, BERT 0.2158) while improving GO accuracy. This pattern reflects the standard precision–diversity trade-off in decoding: more exploratory sampling can surface additional, specific GO candidates that our ontology decoder preserves, at a small cost to phrasing similarity with references. All settings maintain high coverage; $T=0.3$ achieves a marginal peak (0.99), while $T=0.0$ and 0.7 are at 0.98. In practice these differences are negligible. For *best GO performance*, use $T=0.7$ within the agentic cascade. For *highest text fidelity and reproducibility*, $T=0.0$ remains preferred. When reporting main results, we select $T=0.7$ for GO evaluations.

---

**Algorithm 1:** ProtFunAgent Workflow Algorithm

---

**Input** : Schema $x = \{q, a, g, o, f, \mathcal{K}, \mathcal{D}, \mathcal{I}\}$ (sequence $q$, accession $a$, genes $g$, organism $o$, function text $f$, keywords $\mathcal{K}$, domains $\mathcal{D}$, interactions $\mathcal{I}$)

**Input** : Model cascade $\mathcal{M} = \{m_1, \ldots, m_L\}$; attempts per regime $T$; acceptance threshold $\tau$; lexicon $\mathcal{L}$; GO budgets $(K, K_{\mathrm{BP}}, K_{\mathrm{MF}})$

**Output** : Summary $s^\star$; GO leaf set $Y^\star$; hierarchical closure $Y^\uparrow$

---

1   **(1) Homology-RAG**                `// optional but enabled when BLAST is available`
2   $\mathcal{H}(q) \leftarrow \textsc{RetrieveHomologs}(q, k, \theta_{\mathrm{id}}, \theta_E)$
3   $\mathrm{ctx}(q) \leftarrow \textsc{FormatCtx}(\mathcal{H}(q))$             `// do not copy unsupported facts`
4   **(2) Agentic Synth $\rightarrow$ Judge loop**
5   $\mathcal{A}(x) \leftarrow \textsc{BuildAllowlist}(x)$              `// compact lexical anchors`
6   $s^\star \leftarrow \varnothing, best \leftarrow -\infty$
7   **for** $\ell = 1$ **to** $L$ **do**
     `// Normal regime (schema + homology context)`
8     **for** $t = 1$ **to** $T$ **do**
9       $\hat{s} \leftarrow \textsc{Synth}(m_\ell, x, \mathrm{ctx}(q))$
10      $r \leftarrow \textsc{Judge}(m_{\mathrm{J}} \leftarrow m_\ell \text{ or fixed}, x, \hat{s})$        `// ` $r \in \{0, \ldots, 10\}$
11      **if** $r > best$ **then**
12       $s^\star \leftarrow \hat{s}; best \leftarrow r$
13      **end**
14      **if** $r \geq \tau$ **then**
15       **break**                 `// early accept`
16      **end**
17     **end**
18     **if** $best \geq \tau$ **then**
19      **break**
20     **end**
     `// Constrained regime (adds ` $\mathcal{A}(x)$ `)`
21     **for** $t = 1$ **to** $T$ **do**
22      $\hat{s} \leftarrow \textsc{SynthConstrained}(m_\ell, x, \mathrm{ctx}(q), \mathcal{A}(x))$
23      $r \leftarrow \textsc{Judge}(m_{\mathrm{J}}, x, \hat{s})$
24      **if** $r > best$ **then**
25       $s^\star \leftarrow \hat{s}; best \leftarrow r$
26      **end**
27      **if** $r \geq \tau$ **then**
28       **break**
29      **end**
30     **end**
31     **if** $best \geq \tau$ **then**
32      **break**
33     **end**
34   **end**
35   **if** $best < \tau$ **then**
36     $s^\star \leftarrow \textsc{FallbackTemplate}(x)$           `// deterministic, rule-based`
37   **end**
38   **(3) Ontology candidate generation (multi-agent)**
39   $Y_{\mathrm{base}} \leftarrow \textsc{Extract}(s^\star; \mathcal{L})$            `// IDs from baseline summary`
40   $Y_{\mathrm{rich}} \leftarrow \textsc{Extract}(\textsc{GORichProse}(s^\star, x); \mathcal{L})$
41   $Y_{\mathrm{list}} \leftarrow \textsc{MapNamesToIDs}(\textsc{GOFreeList}(x); \mathcal{L})$
42   $\mathcal{A}_{\mathrm{GO}}(x, \mathcal{H}(q)) \leftarrow \textsc{BuildGOAllowlist}(x, \mathcal{H}(q), \mathcal{L})$
43   $Y_{\mathrm{cons}} \leftarrow \textsc{MapNamesToIDs}(\textsc{GOConstrainedSelect}(x, \mathcal{A}_{\mathrm{GO}}); \mathcal{L})$
44   $\tilde{Y} \leftarrow Y_{\mathrm{base}} \cup Y_{\mathrm{rich}} \cup Y_{\mathrm{list}} \cup Y_{\mathrm{cons}}$
45   **(4) Precision-oriented pruning and closure**
46   $Y^\star \leftarrow \textsc{PruneGO}(\tilde{Y}, x, \mathcal{H}(q), \mathcal{L}, K, K_{\mathrm{BP}}, K_{\mathrm{MF}})$
47   $Y^\uparrow \leftarrow \textsc{ExpandAncestors}(Y^\star; \mathcal{L}, \mathrm{aspects} = \{\mathrm{BP}, \mathrm{MF}\})$
48   **return** $s^\star, Y^\star, Y^\uparrow$

---

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
