# PROTFUNAGENT: AGENTIC LLM CASCADES FOR LOW-RESOURCE PROTEIN FUNCTION GAP-FILLING VIA HOMOLOGY RAG AND ONTOLOGY-CONSTRAINED DECODING

## APPENDIX

## A    DATA ACQUISITION, FILTERING, AND SPLITS (FULL DETAILS)

**Source and fields.**    We query the UniProtKB/Swiss–Prot REST API (`https://rest.uniprot.org/uniprotkb/search`) with the pattern `organism_id:$T & reviewed:true`, where $T$ is a taxonomy identifier. For each hit we request only minimal fields to reduce bandwidth and ensure reproducibility: *accession*, *protein name*, *organism ID*, *GO IDs*, *function comment*, and *sequence*. Pagination is handled via cursors (page size 500) with exponential backoff on HTTP 429/5xx responses.

**Uncharacterized heuristic.**    Let $f$ denote the function text, $\mathcal{G}$ the set of GO IDs, and $n(f)$ the tokenized length. We classify an entry as uncharacterized if:

$$\text{UNCHAR}(\text{name}, f, \mathcal{G}) = \mathbf{1} \left[ \begin{array}{l} \text{name contains "uncharacterized", "hypothetical", "putative", or "probable",} \\ \vee \left( n(f) < 25 \wedge (\mathcal{G} = \varnothing \ \vee \ \mathcal{G} \subseteq \{\text{root terms}\} \ \vee \ |\mathcal{G}| \leq 2) \right) \end{array} \right].$$

Root terms are `GO:0008150` (biological process), `GO:0003674` (molecular function), and `GO:0005575` (cellular component). Names are normalized to lowercase and function text is whitespace-normalized.

**Rescue step.**    If a species lacks sufficient labeled examples to satisfy split targets, we attempt to reclassify borderline cases. For proteins initially flagged as uncharacterized, we fetch the full UniProt JSON (`uniprotkb/<acc>.json`) and re-parse both `FUNCTION` comments and GO cross-references. If the entry yields longer function text or non-root GO annotations, it is re-labeled as characterized. This ensures robust per-species coverage without artificially biasing the evaluation.

**CD-HIT clustering.**    To mitigate homolog leakage, we cluster all sequences across species using CD-HIT at $c = 0.60$ identity, following the word-size rule-of-thumb ($n = 4$ at this threshold). We parse the `.clstr` output and enforce cluster-aware splits: entire clusters are assigned to either Dev or Test. Uncharacterized proteins are sampled independently per species. If CD-HIT is unavailable, the script falls back to random splits with an explicit warning.

**Species and caps.**    We target ten species (NCBI TaxIDs): 9606 (human), 559292 (yeast), 83333 (*E. coli*), 3702 (*Arabidopsis*), 7955 (zebrafish), 7227 (fly), 6239 (worm), 10116 (rat), 10090 (mouse), and 4932 (budding yeast). For each, we select up to 500 proteins drawn from $\leq$2000 candidates: 200 Dev (labeled), 200 Test (labeled), 100 Unchar, plus an ablation slice of 100 yeast proteins for fast iteration.

**Reproducibility.**    Exact commands and parameters are provided for transparency:

Listing 1: Command used to create the 10-species benchmark.

```
1  python3 build_splits.py \
2    --out-root data_splits_10species_c60 \
```

```
3    --species 9606 559292 83333 3702 7955 7227 6239 10116 10090 4932 \
4    --per-species 500 \
5    --max-per-species 2000 \
6    --fetch-size 500 \
7    --dev-per-species 200 \
8    --test-per-species 200 \
9    --unchar-per-species 100 \
10   --ablate-yeast 100 \
11   --seed 13 \
12   --cluster-id 0.6
```

**Artifacts.** The pipeline writes: `ids_dev.txt`, `ids_test.txt`, `ids_unchar.txt`, `ids_ablate_yeast.txt`, as well as `metadata.csv` (per-species statistics) and `records.jsonl` (traceable raw records).

**Quality controls.** We release per-species counts after each filter, the proportion classified as uncharacterized, the success rate of the rescue step, and CD-HIT logs. This ensures the benchmark can be audited and replicated.

**Limitations.** REST responses may evolve over time; we log the snapshot date implicitly via timestamps in `records.jsonl`. The uncharacterized heuristic is conservative and may undercount borderline cases in certain taxa. Our ablations analyze sensitivity to the 25-word threshold and the CD-HIT identity cutoff.

## B  DATA, SPLITS, AND PREPROCESSING

**Accession sets.** We formalize three accession sets: **Dev** (labeled, cluster-aware), **Test** (labeled, cluster-aware), and **Unchar** (optional, heuristic-based).

**Schema example.** Each UniProt record is serialized into a compact JSON schema for downstream processing. An example (truncated) is shown below:

Listing 2: Schema JSON exemplar (truncated).

```
1    {
2      "accession": "P25623",
3      "gene": ["SYP1"],
4      "organism": "Saccharomyces cerevisiae",
5      "length_aa": 829,
6      "function_text": "Plays a role in ...",
7      "go": {
8        "BP": [{"term": "endocytosis"}],
9        "MF": [{"term": "protein binding"}],
10       "CC": []
11     },
12     "keywords": ["Coated vesicles"],
13     "interactions": [{"partner": "ENT1", "accession": "Q..." }],
14     "domains": [{"name": "BAR domain", "range": "20-240"}],
15     "sequence_fasta": ">P25623\nMTHQ..."
16   }
```

## C  ONTOLOGY LEXICON AND MAPPING

**Graph source and aspects.** We parse `go-basic.obo` with obonet, restricting to Biological Process (BP) and Molecular Function (MF). The resulting lexicon $\mathcal{L}$ contains: (i) names and synonyms → IDs (lowercased), and (ii) adjacency via *is_a* and *part_of* relations.

Table 1: Ontology lexicon and mapping statistics (example values).

| Metric | Value (BP+MF) |
|---|---|
| Unique GO phrases | 43,512 |
| GO IDs covered | 25,781 |
| Mean parents/child | 1.83 |
| Name→id entries | 47,605 |

**Synonym normalization.** Phrases are normalized by lowercasing, collapsing whitespace, and stripping punctuation. Regex-based extraction compiles batches of boundary-guarded patterns to avoid regex-size limits.

Listing 3: Phrase normalization and matching (sketch).

```
1  normalize(p): lowercase -> collapse spaces -> strip punctuation
2  compile_candidate_regex(phrases): chunk -> escape -> boundary-guarded OR
3  extract_terms(text,L): iterate regex batches; de-duplicate by GO id
```

**Name→id map.** We additionally build a mapping $\mathcal{M}$ using make_go_map.py. This merges canonical names from id2term with all synonyms in phrase2id, producing a dense dictionary of normalized names to IDs. $\mathcal{M}$ is used for: (i) mapping free lists of GO names to IDs, (ii) schema evidence checks, and (iii) evaluation alignment.

**Statistics.** We report: #unique phrases, #GO IDs (BP/MF), mean parents per child, and size of $\mathcal{M}$ (name→id entries).

## D HOMOLOGY-RAG IMPLEMENTATION DETAILS

**BLAST database.** We use SwissProt (*date: yyyy-mm-dd*); run makeblastdb once:

Listing 4: Building a BLASTP database (example).

```
1  makeblastdb -in uniprot_sprot.fasta -dbtype prot -out uniprot_sprot
```

**Retrieval parameters.** Defaults: $k{=}3$, $\theta_{\mathrm{id}}{=}30\%$, $\theta_E{=}10^{-5}$. We select unique accessions by (best e-value, then identity), discard the query accession, and inlinedly fetch UniProt FUNCTION text for each hit.

**Context formatting.** We render homologs as a bullet list with identity and E-value. Include an example block here (anonymized).

## E AGENTIC LOOP: PROMPTS AND CASCADE POLICY

**Prompt library overview.** We store each prompt template explicitly; below we reproduce the core variants.

> **Synth (Normal)**
>
> You are a scientific writer. Write ONE paragraph ($\leq$120 words) summarizing protein function from the JSON. Be precise and do NOT invent facts. Prefer canonical phrases present in the input.
> **Input:** Schema JSON + optional homolog context **Output:** Concise summary paragraph

**Synth (GO-rich)**

Using ONLY the JSON, write ONE compact paragraph ($\leq$120 words) that enumerates as many Biological Process (BP) and Molecular Function (MF) terms as possible. Do NOT invent facts.

**Synth (Constrained)**

Using ONLY the JSON fields, produce $\leq$90 words. Use ONLY words/phrases from the ALLOWED TERMS and glue words. Prefer BP, then MF, then domains/interactions. Do NOT invent facts.

**GO List Extractor**

Extract ALL distinct GO term NAMES present in the JSON for BP and MF. Output: a single line, comma-separated list. If none, output EMPTY.

**GO-Constrained Selector**

From the JSON, select the MOST RELEVANT GO terms, choosing ONLY from the allowlist. Output: comma-separated list of terms.

**Judge**

You are a strict fact-checker. Flag any spans not supported by JSON values. Rewrite if needed. Output strictly:
SCORE: <1–10> SUMMARY: <final one-paragraph summary $\leq$100 words>

**Fallback (Rule-based)**

If no candidate exceeds threshold $\tau$, output a slot-filled template: (gene, organism, top BP/MF terms, domains, interactors).

## F  GO SELECTION AND PRUNING — FULL SPECIFICATION

**Candidate sources.** $Y_{\text{base}} = \text{Extract}(s^\star; \mathcal{L})$, $Y_{\text{rich}}, Y_{\text{list}}, Y_{\text{cons}}$ from the three GO agents.

**Support function.** Each predicted term $g$ receives $w(g) = 2 \cdot \mathbf{1}[g$ in schema BP/MF$] + \mathbf{1}[g \in Y_{\text{cons}}]$ (capped at 2). Optionally require $w(g) > 0$.

**Specificity preference.** Define depth $d(g)$ as the size of the ancestor closure (or a graph-theoretic depth); sort by $(w(g)\downarrow, d(g)\downarrow)$, then drop any ancestor when a child is present.

**Per-aspect quotas.** Apply $(K, K_{\text{BP}}, K_{\text{MF}})$ to form $Y^\star$; compute closure $Y^\uparrow$ for evaluation.

---

**Algorithm 1:** Precision-Oriented GO Pruning (full)

---

**Input:** $\tilde{Y}$, schema $x$, homology $\mathcal{H}(q)$, lexicon $\mathcal{L}$, budgets $(K, K_{\mathrm{BP}}, K_{\mathrm{MF}})$
**Output:** $Y^{\star}$
**1 for** $g \in \tilde{Y}$ **do**
**2** $\quad w(g) \leftarrow 2 \cdot \mathbf{1}[g \in \mathrm{schemaBP/MF}] + \mathbf{1}[g \in Y_{\mathrm{cons}}]$
**3 end**
**4 if** REQUIRESUPPORT **then**
**5** $\quad \tilde{Y} \leftarrow \{g \in \tilde{Y} : w(g) > 0\};$ **if** $\tilde{Y} = \varnothing$ **then**
**6** $\quad\quad$ **return** $\varnothing$
**7** $\quad$ **end**
**8 end**
**9** compute $d(g)$ via closure size; order $\hat{Y}$ by $(w(g){\downarrow}, d(g){\downarrow})$
**10** $Y' \leftarrow \varnothing;$ **for** $g \in \hat{Y}$ **do**
**11** $\quad$ **if** $\nexists h \in Y'$ *s.t.* $g \in \mathrm{Anc}(h)$ **then**
**12** $\quad\quad$ $Y' \leftarrow Y' \cup \{g\}$
**13** $\quad$ **end**
**14 end**
**15** $Y^{\star} \leftarrow$ apply aspect quotas and total cap to $Y'$
**16 return** $Y^{\star}$

---

# G   BASELINE METHODS

## G.1   SINGLE-LLM (FREE-FORM) BASELINE AND RUNNER

This baseline isolates the contribution of a *single* backbone LLM that writes a free-form function summary from the parsed UniProt schema, without cascades, judges, or retrieval. GO terms are then decoded from the generated text using the same ontology tools as in our main pipeline. A lightweight runner orchestrates generation and evaluation consistently across data splits.

**Setting.** For each accession $a$ with parsed schema $\sigma(a)$ (genes, organism, UniProt FUNCTION text, GO cross-refs, domains, etc.), we prompt a backbone LLM $\mathcal{P}_{\theta}$ to produce a concise free-form summary

$$s^{\star}(a) \;=\; \arg\max_{s \in \mathbb{T}} \; \mathcal{P}_{\theta}\big(s \,\big|\, \sigma(a)\big),$$

under a shortness/style instruction but *without* lexical fencing, homology context, or self-judging. This captures the "pure LLM" behavior on our task.

**Ontology decoding.** From $s^{\star}(a)$ we extract a candidate set of GO identifiers by lexicon matching (names + synonyms) and map-to-ID:

$$\tilde{Y}(a) \;=\; \mathrm{Extract}\big(s^{\star}(a); \mathcal{L}\big),$$

followed by aspect-aware pruning with budgets $(k_{\mathrm{BP}}, k_{\mathrm{MF}})$ and a total cap $k_{\mathrm{tot}}$:

$$Y^{\mathrm{leaf}}(a) \;=\; \big(\tilde{Y}(a) \cap \mathrm{BP}\big)_{[:k_{\mathrm{BP}}]} \;\cup\; \big(\tilde{Y}(a) \cap \mathrm{MF}\big)_{[:k_{\mathrm{MF}}]}, \qquad \big|Y^{\mathrm{leaf}}(a)\big| \leq k_{\mathrm{tot}}.$$

For hierarchical metrics we compute ancestor closure in the GO DAG $\mathcal{G}_{\mathrm{GO}}$ (restricted to BP/MF):

$$Y^{\uparrow}(a) \;=\; \mathrm{Ancestors}\big(Y^{\mathrm{leaf}}(a); \mathcal{G}_{\mathrm{GO}}\big) \;\cup\; Y^{\mathrm{leaf}}(a).$$

**Outputs and evaluation.** Per accession we return the triple $\big(s^{\star}(a), Y^{\mathrm{leaf}}(a), Y^{\uparrow}(a)\big)$. The runner executes this pipeline split-by-split (dev, test, unchar, etc.) and evaluates with the same metrics used throughout the paper:

- **Text:** ROUGE-L and BERTScore-F1 computed on non-empty pairs after clamping reference length.
- **GO flat (leaves):** micro/macro P/R/F1 on $Y^{\mathrm{leaf}}(a)$.
- **GO hierarchical:** micro/macro P/R/F1 on $Y^{\uparrow}(a)$.

- **Ontology consistency:** graded score in $[0, 1]$ reflecting ancestor completeness (roots disallowed).

- $K$**-sweep PR:** micro P/R/F1 as we truncate the predicted order to $K$ (fixed points per split).

The runner ensures reproducibility by (i) consuming the canonical parsed schemas, (ii) writing per-split predictions to a fixed location, and (iii) invoking the same evaluator and $K$-sweep used for the other baselines and our full agentic system.

**Why this baseline matters.** This variant represents the best case for a *single* generative model that reads only the schema and writes a summary in one pass. It tests whether fluent generation alone can (i) align with gold-standard text and (ii) implicitly surface ontology terms that the lexicon can capture. Unlike the constrained baseline, it has no lexical guardrails; unlike the homology baseline, it has no sequence-level evidence; and unlike ProtFunAgent, it has no self-verification, cascades, or ontology-constrained selection at generation time.

**Limitations.** Free-form generation is prone to (i) *hallucination*—phrases that are fluent yet unsupported by the schema; (ii) *phrase drift*: valid biology phrased in ways that miss lexicon matches, harming GO recall; and (iii) *lack of calibration*: no judge or acceptance threshold. Empirically, we observe that while a single-LLM baseline can attain reasonable GO performance on familiar patterns, its text faithfulness and structured recall lag behind our agentic, homology-aware, and ontology-constrained pipeline.

### G.2 HOMOLOGY-ONLY BASELINE

A long-standing heuristic in protein function prediction is homology-based transfer, where annotations are inferred from close sequence neighbors. We implemented a homology-only baseline that leverages BLASTP alignment against the SwissProt database, followed by ontology-aware term extraction and pruning.

**Method.** Given a query protein sequence $q$, we retrieve its top-$k$ homologs

$$\mathcal{H}(q) = \{h_1, h_2, \ldots, h_k\},$$

filtered by sequence identity $\rho(h_j, q) \geq \theta_{\text{id}}$ and alignment e-value $E(h_j, q) \leq \theta_E$. From the associated UniProt entries of these homologs, we extract functional text descriptions $f(h_j)$. Each description is scanned for lexical matches to Gene Ontology terms using a precompiled lexicon $\mathcal{L}$ of names and synonyms. This yields a candidate set of GO identifiers:

$$\tilde{Y}(q) = \bigcup_{j=1}^{k} \text{Extract}(f(h_j); \mathcal{L}).$$

**Ontology Expansion and Pruning.** To ensure hierarchical coverage, we expand the predictions by computing the ancestor closure of $\tilde{Y}(q)$ in the GO DAG:

$$Y^{\uparrow}(q) = \text{ExpandAncestors}\big(\tilde{Y}(q); \mathcal{G}\big),$$

where $\mathcal{G}$ denotes the GO ontology graph with edges defined by `is_a` and `part_of` relations.

We then prune this set to maintain interpretability and avoid generic terms. Specifically, we allocate a budget of $k_{\text{BP}}$ terms to biological process and $k_{\text{MF}}$ terms to molecular function:

$$Y^{\star}(q) = \text{Prune}\big(Y^{\uparrow}(q), k_{\text{BP}}, k_{\text{MF}}, k_{\text{tot}}\big).$$

**Output.** The final output is a structured summary and ontology-consistent GO term set:

$$\hat{y}(q) = \big(Y^{\star}(q), \text{Summary}(Y^{\star}(q))\big),$$

where $\text{Summary}(\cdot)$ produces a short textual rationale listing the predicted functions.

This reflects the conventional paradigm of annotation transfer by sequence similarity, augmented here by an ontology lexicon for consistency. Its strengths lie in high precision when close homologs exist, but limitations include poor coverage for low-identity queries, susceptibility to annotation propagation errors, and lack of free-text rationalization beyond templated summaries. We therefore treat it as a lower-bound comparator against which agentic LLM pipelines can demonstrate improvements in generalization and interpretability.

### G.3 TEMPLATE BASELINE

This baseline produces a fully deterministic annotation directly from the parsed UniProt record (schema) without invoking any LLM. It serves as a strong non-neural reference because it exploits the curated fields already present in many entries.

**Inputs.** From the schema we read: accession $a$, primary/synonym gene names $\mathcal{G}$, organism $o$, free-text functional note $u$ (if present), and GO seed names per aspect $\mathcal{S}_{\mathrm{BP}}$ and $\mathcal{S}_{\mathrm{MF}}$ extracted from the record's GO cross-references.

**Deterministic summary.** We form a short natural-language synopsis by stitching fixed clauses from the schema:

$$\mathrm{Summary}(a) = \underbrace{\mathrm{NameBlock}(a, \mathcal{G}, o)}_{\text{identity}} + \underbrace{\mathrm{BP\text{-}clause}(\mathcal{S}_{\mathrm{BP}})}_{\text{process}} + \underbrace{\mathrm{MF\text{-}clause}(\mathcal{S}_{\mathrm{MF}})}_{\text{function}},$$

where each clause lists up to a small, fixed number of unique terms (e.g., top 4 BP and top 3 MF after de-duplication). This yields stylized but fluent text with zero stochasticity.

**Ontology mapping and quotas.** We map each GO seed name to its identifier using a lexicon $\mathcal{L}$ of canonical names and synonyms:

$$\tilde{Y} = \{\, \mathrm{map}(s; \mathcal{L}) \ : \ s \in \mathcal{S}_{\mathrm{BP}} \cup \mathcal{S}_{\mathrm{MF}} \,\}.$$

To avoid aspect imbalance and overly long lists, we apply simple per-aspect budgets

$$Y^{\mathrm{leaf}} = \big(\tilde{Y} \cap \mathrm{BP}\big)_{[:k_{\mathrm{BP}}]} \cup \big(\tilde{Y} \cap \mathrm{MF}\big)_{[:k_{\mathrm{MF}}]}, \qquad |Y^{\mathrm{leaf}}| \leq k_{\mathrm{tot}}.$$

**Hierarchical closure.** For evaluation that requires ontology consistency, we compute the ancestor closure under the GO DAG $\mathcal{G}_{\mathrm{GO}}$ using `is_a`/`part_of`:

$$Y^{\uparrow} = \mathrm{Ancestors}\big(Y^{\mathrm{leaf}}; \mathcal{G}_{\mathrm{GO}}\big) \cup Y^{\mathrm{leaf}},$$

optionally restricted to aspects $\{\mathrm{BP}, \mathrm{MF}\}$.

**Outputs.** The baseline returns the pair

$$\hat{y}_{\mathrm{tmpl}} = \Big(\mathrm{Summary}(a),\ Y^{\mathrm{leaf}},\ Y^{\uparrow}\Big),$$

i.e., a templated summary plus leaf and hierarchical GO sets.

The template baseline has three desirable properties: (i) *determinism and speed*—no inference cost, making it a robust fallback; (ii) *high precision on curated entries*—when GO seeds exist, mapping is exact; (iii) *ontology-faithfulness* via explicit closure. Its limitations mirror schema sparsity: when records are incomplete or lack GO seeds, recall is intrinsically capped, and the prose cannot generalize beyond what is present in the schema. Unlike our agentic pipeline, it cannot synthesize new evidence from homologs nor resolve conflicting hints across fields. Consequently, it provides a conservative lower bound for both text metrics (ROUGE/BERTScore) and GO F1, against which gains from homology-RAG, synthesis–judging, and ontology-constrained decoding can be quantified.

### G.4 SCHEMA–GO–COPY BASELINE

This baseline measures the performance obtainable by *directly copying* GO terms already present in the parsed UniProt record, without any generation, retrieval, or inference. It therefore acts as an "oracle–seed" reference: whenever the record carries curated GO cross-references, those are mapped and scored; otherwise the prediction is empty.

**Inputs.** From the parsed schema $\sigma(a)$ of accession $a$ we read the per-aspect GO name lists

$$\mathcal{S}_{\mathrm{BP}}(a), \quad \mathcal{S}_{\mathrm{MF}}(a),$$

and a GO lexicon $\mathcal{L}$ containing canonical names/synonyms and a parent mapping over the GO DAG $\mathcal{G}_{\mathrm{GO}}$.

**Name → ID mapping.** We convert seed names to identifiers using $\mathcal{L}$:

$$\tilde{Y}(a) = \big\{ \mathrm{map}(s; \mathcal{L}) \ : \ s \in \mathcal{S}_{\mathrm{BP}}(a) \cup \mathcal{S}_{\mathrm{MF}}(a) \big\}.$$

No lexical expansion, paraphrase, or generation is performed; if a seed lacks a lexicon entry, it is dropped.

**Leaf set and hierarchical closure.** Because the intent is to mirror the record, we retain all available leaves (no budget limits):

$$Y^{\mathrm{leaf}}(a) = \tilde{Y}(a).$$

For hierarchical metrics and ontology checks, we compute the ancestor closure in $\mathcal{G}_{\mathrm{GO}}$ (restricted to BP/MF):

$$Y^{\uparrow}(a) = \mathrm{Ancestors}\big(Y^{\mathrm{leaf}}(a); \mathcal{G}_{\mathrm{GO}}\big) \cup Y^{\mathrm{leaf}}(a).$$

**Outputs.** We emit a minimalist textual tag (for bookkeeping) together with the copied GO sets:

$$\hat{y}_{\mathrm{copy}}(a) = \big(\text{"summary: copied from record"}, \ Y^{\mathrm{leaf}}(a), \ Y^{\uparrow}(a)\big).$$

Strengths: (i) *Upper bound on schema-recoverable labels*: whenever curated GO is present, this baseline can match it exactly; (ii) *Deterministic and fast*: there is no stochasticity or model inference; (iii) *Ontology-faithful* after closure.

Limitations: (i) *No generalization*: it cannot infer functions absent from the record, making recall zero for entries without GO seeds; (ii) *No textual evidence*: the "summary" is non-informative and unsuitable for curatorial workflows; (iii) *No reconciliation*: conflicting or overly generic seeds (e.g., near-root terms) are not filtered by support or specificity. Consequently, Schema–GO–Copy serves as a conservative reference to quantify the added value of our learned components, homology-guided retrieval, controlled generation, and agentic Synth → Judge cascades over merely echoing what is already curated.

G.5    EXTRACTIVE BASELINE

The extractive baseline is a non-generative reference that (i) truncates the record's curated function text to a fixed budget and (ii) derives GO terms by lexicon matching on that same text, followed by ontology-aware post-processing. It represents an upper bound on what can be recovered from the *given* UniProt note without synthesis.

**Inputs.** From the parsed schema we use only the free-text function description $u \in \mathbb{T}$ and a GO lexicon $\mathcal{L}$ (canonical names, synonyms, and parent map).

**Extractive summary.** We form a deterministic synopsis by hard-truncation at the word level:

$$\mathrm{Summ}_{\mathrm{ext}}(u; W) = \mathrm{Join}\Big( \big(\mathrm{Tok}(u)\big)_{[:W]} \Big),$$

where $\mathrm{Tok}(\cdot)$ tokenizes on whitespace and $W$ is a small budget (e.g., $W{=}120$). This preserves curator phrasing while eliminating verbosity.

**Lexicon match for GO candidates.** We scan the lower-cased text for lexicon phrases and map to GO identifiers:

$$\tilde{Y} = \Big\{ \mathrm{map}(p; \mathcal{L}) \ \Big| \ p \in \mathrm{Phrases}(\mathcal{L}), \ p \subseteq \mathrm{lower}(u) \Big\}.$$

This yields a high-precision but recall-limited candidate set tied strictly to what appears in the note.

**Aspect-aware pruning and hierarchical closure.** To keep predictions concise and balanced across aspects, we apply simple quotas:

$$Y^{\mathrm{leaf}} = \big(\tilde{Y} \cap \mathrm{BP}\big)_{[:k_{\mathrm{BP}}]} \ \cup \ \big(\tilde{Y} \cap \mathrm{MF}\big)_{[:k_{\mathrm{MF}}]}, \qquad |Y^{\mathrm{leaf}}| \leq k_{\mathrm{tot}}.$$

For hierarchical evaluation and ontology faithfulness, we compute ancestor closure on the GO DAG $\mathcal{G}_{\mathrm{GO}}$ using is_a/part_of relations, restricted to BP/MF:

$$Y^{\uparrow} = \mathrm{Ancestors}\big(Y^{\mathrm{leaf}}; \mathcal{G}_{\mathrm{GO}}\big) \ \cup \ Y^{\mathrm{leaf}}.$$

**Outputs.** The baseline returns

$$\hat{y}_{\text{ext}} \;=\; \Big(\text{Summ}_{\text{ext}}(u; W),\; Y^{\text{leaf}},\; Y^{\uparrow}\Big),$$

i.e., a truncated extractive summary and corresponding leaf/hierarchical GO sets derived solely from the curated text.

This method leverages the strongest available prior—the curator's own wording—yielding (i) *deterministic, high-precision* GO matches when canonical phrases occur, and (ii) *faithful* summaries that reflect existing annotations. However, its recall is inherently capped: any function not explicitly mentioned in $u$ cannot be recovered, and synonym/phrase mismatches degrade coverage even with a rich lexicon. Moreover, unlike the agentic pipeline, it neither integrates homology evidence nor reconciles conflicting cues across fields. We therefore use the extractive baseline as a conservative reference that isolates the contribution of *synthesis* (LLM-generated text) and *retrieval* (Homology-RAG) in ProtFunAgent.

G.6 CONSTRAINED (ALLOWLIST) BASELINE

This baseline generates a concise function synopsis by prompting an LLM under a hard lexical constraint (an *allowlist*) derived deterministically from the record. GO terms are then extracted from the constrained prose and post-processed with ontology rules. The goal is to isolate the effect of *controlled generation* without external retrieval or free-form paraphrase.

**Inputs.** From the parsed UniProt record (schema) we read identifier $a$, gene/name tokens $\mathcal{G}$, organism token $o$, the function note $u$, and GO seed names per aspect $\mathcal{S}_{\text{BP}}, \mathcal{S}_{\text{MF}}$. We also assume a GO lexicon $\mathcal{L}$ containing canonical names, synonyms, and a parent map over the GO DAG $\mathcal{G}_{\text{GO}}$.

**Allowlist construction.** We deterministically assemble a finite vocabulary $\mathcal{V}$ from schema tokens:

$$\mathcal{V} \;=\; \underbrace{\text{Tok}(\mathcal{G}) \cup \{o\}}_{\text{identity}} \;\cup\; \underbrace{\text{Tok}(u)}_{\text{curator text}} \;\cup\; \underbrace{\bigcup_{s \in \mathcal{S}_{\text{BP}} \cup \mathcal{S}_{\text{MF}}} \text{Tok}(s)}_{\text{GO seeds}},$$

followed by normalization and length filtering (e.g., alphanumerics, $|w| \le 30$). This yields a record-specific vocabulary that (i) carries factual anchors and (ii) strongly curbs hallucination.

**Constrained summarization.** We query an LLM with a prompt that (i) instructs a short ($\le 120$ words) factual summary, (ii) forbids inventing facts, and (iii) restricts the output token set to $\mathcal{V}$ (plus numerals and punctuation). Denote the resulting summary by

$$s_{\mathcal{V}}^{*} \;=\; \arg\max_{s \in \mathbb{T}} \; \mathcal{P}_{\theta}(s \mid \text{schema}) \quad \text{s.t.} \quad \text{Tok}(s) \subseteq \mathcal{V} \cup \Sigma_{\text{num/punc}}.$$

This "lexically fenced" decoding preserves fluency while acting as a content filter aligned to the schema.

**GO extraction, pruning, closure.** We match lexicon phrases inside the constrained summary to generate candidate GO identifiers:

$$\tilde{Y} \;=\; \big\{ \text{map}(p; \mathcal{L}) \;:\; p \in \text{Phrases}(\mathcal{L}),\; p \subseteq \text{lower}(s_{\mathcal{V}}^{*}) \big\}.$$

To ensure concision and aspect balance, we apply budgets (BP first, then MF):

$$Y^{\text{leaf}} \;=\; \big(\tilde{Y} \cap \text{BP}\big)_{[:k_{\text{BP}}]} \;\cup\; \big(\tilde{Y} \cap \text{MF}\big)_{[:k_{\text{MF}}]}, \qquad |Y^{\text{leaf}}| \le k_{\text{tot}}.$$

For hierarchical consistency we compute ancestor closure on the GO DAG (restricted to BP/MF):

$$Y^{\uparrow} \;=\; \text{Ancestors}\big(Y^{\text{leaf}}; \mathcal{G}_{\text{GO}}\big) \cup Y^{\text{leaf}}.$$

**Outputs.** The baseline returns

$$\hat{y}_{\text{constr}} \; = \; \bigl(s_{\mathcal{V}}^{*}, \; Y^{\text{leaf}}, \; Y^{\uparrow}\bigr),$$

i.e., a concise allowlisted summary together with ontology-consistent GO predictions.

The constrained baseline offers three advantages over purely extractive or free-form generation: (i) *faithfulness*—the allowlist tightly couples the summary to schema evidence, sharply reducing hallucination; (ii) *coverage beyond exact string match*—the LLM can recombine allowed tokens, surfacing lexicon phrases that a strict extractive matcher might miss; and (iii) *compatibility with ontology post-processing*. Its chief limitations are sensitivity to allowlist quality (over-restrictive vocabularies can harm recall) and lack of external grounding (no homology or literature retrieval). In our results, it serves as a mid-point baseline: stronger than extractive when schema text is rich, but generally outperformed by the agentic pipeline with homology-augmented prompts and multi-source ontology decoding.

### G.7 RANDOM–GO BASELINE

As a lower-bound control we include a purely random baseline that samples GO terms directly from the lexicon without regard to sequence or schema content. Specifically, we uniformly shuffle the set of BP and MF identifiers, then select up to $k_{\text{BP}}$ and $k_{\text{MF}}$ terms (capped at $k_{\text{tot}}$ overall). The chosen identifiers

$$Y^{\text{leaf}} \; = \; \text{Sample}\bigl(\mathcal{L}_{\text{BP}}, k_{\text{BP}}\bigr) \; \cup \; \text{Sample}\bigl(\mathcal{L}_{\text{MF}}, k_{\text{MF}}\bigr)$$

are then closed under ontology ancestors

$$Y^{\uparrow} \; = \; \text{Ancestors}\bigl(Y^{\text{leaf}}; \mathcal{G}_{\text{GO}}\bigr) \; \cup \; Y^{\text{leaf}}.$$

Each record is paired with a generic placeholder summary ("Random GO control"), ensuring that evaluation focuses exclusively on GO prediction quality.

This provides a sanity check: (i) it establishes a chance-level expectation for recall and precision under ontology-constrained random sampling; (ii) it verifies that higher-performing methods derive signal from data rather than from lexicon size or metric artifacts. Unsurprisingly, Random–GO yields very low F1 and consistency, but it helps contextualize the magnitude of improvements observed with homology-aware, constrained, or agentic approaches.

## H EVALUATION PROTOCOL: FULL DETAILS

**Ground-truth mapping.** GO *names* are mapped to IDs using a consolidated name→id table anchored in $\mathcal{L}$; IDs are normalized to GO:NNNNNNN.

**Hierarchical vs flat metrics.** We report micro/macro $P, R, F_1$ for hierarchical ($Y^{\uparrow}$ vs $G$) and flat ($Y^{\star}$ vs $G$) sets. Text metrics: ROUGE-L (stemming on), BERTScore-F1 (baseline rescaling on unless noted).

**Ontology consistency and empty handling.** OC is graded as $1 - \frac{|\text{Anc}(Y^{\star}) \setminus Y^{\star}|}{|\text{Anc}(Y^{\star})|}$; score 0 if $Y^{\star} = \varnothing$ or root terms occur. Text scoring ignores empty candidate/reference pairs; counts reported.

$K$**-sweep.** Top-$K$ truncation respects predicted order; $K \in \{4, 6, 8, 10, 12\}$.

**Reproducibility knobs.** Document -drop-empty, -no-bert-rescale, -max-ref-words, and any random seeds.

## I  SYSTEM, COMPUTE, AND RUNTIME

Table 2: Environment and runtime for the ablation run (250 proteins).

| Item | Value |
|---|---|
| GPU | NVIDIA Tesla P40 (24 GB), Driver 535.261.03, CUDA 12.2 |
| CPU | (fill with lscpu output, e.g., Intel Xeon, N cores) |
| RAM | (fill with free -h, e.g., 64 GB) |
| BLAST DB | SwissProt (uniprot_sprot), size: (fill from du -sh) |
| Homology filters | identity $\geq 30\%$, $E$-value $\leq 10^{-5}$, top-$k = 3$ unique hits |
| LLM backend | Ollama (keep-alive 30 min), temperature 0.0 (unless noted) |
| Cascade | synth $\rightarrow$ judge (same model if JUDGE_MODEL unset); constrained GO decoding enabled |
| Total runtime | $\approx 1200$ s for 250 accessions |
| Median time / accession | $\approx 4.8$ s (1200/250) |
| #Accessions per split | Dev (tuning): fill, Test (reporting): fill, Ablation: 250 |

## J  QUALITATIVE CASE STUDIES: AGENTIC VS. SINGLE-LLM ON UNCHARACTERIZED PROTEINS

### Case 1: P0A6A3 (*E. coli* K-12; gene ackA)

**Agentic summary (ours).**

This protein, encoded by the ackA gene in *Escherichia coli* (K-12), forms acetyl phosphate from acetate and ATP (acetate kinase activity) and contributes to ATP production during anaerobic growth. *(Judge removed an unsupported sentence about the reverse reaction in an earlier draft.)*

**Single-LLM summary (no judge/constraints).**

This enzyme, encoded by the ackA gene in *Escherichia coli* K12, catalyzes the formation of acetyl phosphate from acetate and ATP. It also participates in the reverse reaction and is involved in the synthesis of most ATP formed catabolically during anaerobic growth. The enzyme's activity is associated with magnesium ion binding, zinc ion binding, and acetate kinase activity, and it is localized to both the cytosol and membrane compartments.

**GO predictions (ours).** GO:0008776 acetate kinase activity , GO:0005524 ATP binding ,
GO:0019413 acetyl–CoA biosynthesis , GO:0019542 propionate biosynthesis

**GO predictions (single-LLM).** GO:0000287 $Mg^{2+}$ binding , GO:0004096 catalase activity ,
GO:0004314 ACP S–malonyltransferase , GO:0031578 spindle checkpoint ,
GO:0070914 UV excision repair , GO:0009058 biosynthetic process , GO:0040007 growth

**Analysis.** The single-LLM injects multiple *off-pathway* GO terms (**catalase**, **malonyltransferase**, **UV repair**, **spindle checkpoint**), none of which are consistent with acetate kinase function. The agentic pipeline's *judge* removes unsupported claims; the *ontology decoder* selects the correct MF and specific biosynthesis processes, yielding an ontology-faithful, mechanism-centered summary.

### Case 2: O97422 (GlcAT-I; glycosaminoglycan biosynthesis)

**Agentic summary (ours).**

Galactosylgalactosylxylosylprotein 3-$\beta$-glucuronosyltransferase I participates in L2/HNK-1 epitope formation and is involved in chondroitin and heparan sulfate proteoglycan biosynthesis at the Golgi membrane.
*(Judge trimmed over-claims about substrate breadth.)*

**Single-LLM summary (no judge/constraints).**

The Galactosylgalactosylxylosylprotein 3-beta-glucuronosyltransferase I (GlcAT-I) enzyme is involved in the biosynthesis of L2/HNK-1 carbohydrate epitope on glycolipids and glycoproteins, specifically targeting Gal-$\beta$1,3-Gal-$\beta$1,4-Xyl substrates. This process occurs within the Golgi membrane. The enzyme's activity is also linked to carbohydrate metabolic processes, including chondroitin sulfate proteoglycan and heparan sulfate proteoglycan biosynthesis.

**GO predictions (ours).** GO:0050650 chondroitin sulfate PG biosynthesis , GO:0015012 heparan sulfate PG biosynthesis , GO:0009058 biosynthetic process

**GO predictions (single-LLM).** GO:0000271 polysaccharide biosynthesis , GO:0004096 catalase activity , GO:0004586 ornithine decarboxylase , GO:0015012 heparan sulfate PG biosynthesis , GO:0031578 spindle checkpoint , GO:0043879 glycolate transporter , GO:0009058 biosynthetic process , GO:0005975 carbohydrate metabolism

**Analysis.** Although the single-LLM mentions the right pathways, it adds **catalase**, **ornithine decarboxylase**, **spindle checkpoint**, and a **glycolate transporter**, which are biologically implausible for GlcAT-I. The agentic pipeline restrains outputs to *proteoglycan biosynthesis* with correct cellular context via constrained decoding.

**Case 3: V6CLJ5** (*C. elegans* Lag-1/CSL; Notch axis)

**Agentic summary (ours).**

Lag-1 is a CSL-family regulator acting in `lin-12`/Notch and `glp-1`/Notch signaling to control cell–cell communication, germline stem cell division, and reproductive system development.

**Single-LLM summary (no judge/constraints).**

The Lag-1 protein is a transcriptional regulator central to `lin-12`/Notch and `glp-1`/Notch signaling pathways. It binds specific DNA sequences and can act as repressor or activator with Notch proteins. It is involved in germline stem cell division, reproductive system development, and left-right asymmetry induction.

**GO predictions (ours).** GO:0005112 Notch binding , GO:0007219 Notch signaling , GO:0000978 Pol II cis-reg. DNA binding , GO:0045944 positive regulation of transcription by Pol II

**GO predictions (single-LLM).** GO:0003904 photolyase activity , GO:0004096 catalase activity , GO:0004586 ornithine decarboxylase , GO:0006351 DNA-templated transcription , GO:0007154 cell communication , GO:0007165 signal transduction , GO:0007219 Notch signaling , GO:0007601 visual perception

**Analysis.** The single-LLM correctly references Notch signaling but introduces **photolyase**, **catalase**, **ornithine decarboxylase**, and **visual perception**, none expected for CSL/Notch TFs. The agentic pipeline surfaces specific *transcriptional* and *Notch-binding* GO terms via ontology-aware selection, avoiding generic or irrelevant functions.

## K  EXTENDED ERROR ANALYSIS

Qualitative examples: (1) Synth hallucination caught by Judge; (2) GO ancestor inconsistency fixed by pruning; (3) edge cases with synonym mapping.

---

**Example.** Input schema (abridged) → Synth candidate → Judge score → Final summary; GO candidates and pruned set with reasons.

---

Figure 1: Worked example with intermediate artifacts.

## L  REPRODUCIBILITY CHECKLIST

- Data: accession lists per split (URLs or file hashes), UniProt snapshot date.
- Lexicon: OBO version and date; saved $\mathcal{L}$ checksum.
- Homology-RAG: BLASTP version, DB snapshot date, parameters.
- Models: exact model IDs/tags, temperature, token caps, judge threshold $\tau$, cascade order.
- Seeds and non-determinism notes; retry policy.
- Command lines for all main runs and ablations.