# OpenReview forum: "ProtFunAgent: Agentic LLM Cascades for Low-Resource Protein Function Gap-Filling via Homology RAG and Ontology-Constrained Decoding"
_ICLR.cc/2026/Conference — ICLR 2026 Conference Withdrawn Submission_

### Official Review · Reviewer_vcH7 · 2025-10-21

**Soundness:** 2
**Presentation:** 2
**Contribution:** 2
**Rating:** 2
**Confidence:** 4

**Summary:**

The paper proposes ProtFunAgent, an LLM-based agentic pipeline for protein function annotation. It introduces an RAG module to incorporate information from homogeneous proteins and multiple decoding strategies to extract hierachial GO annotations. Moreover, the framework introduces a multi-step proposer-judger workflow to generate and refine high-quality function descriptions. The work develops a benchmark derived from UniProt with several huerestic-based or LLM-based baselines and performs analysis with comprehensive metrics concerning GO accuracy, consistency, and text fidelity, successfully demonstrating the effectiveness of the proposed agentic pipeline.

**Strengths:**

- The paper address a critical problem of existing LLMs in protein function interpretation: lack of biological grounding evidence from homologous proteins and awareness of GO hierachy.
- A diverse set of evaluation metrics are introduced to comprehensively assess the accuracy, quality, and consistency of the generated annotations.
- The experiment analysis on LLM variants, ensembling, and decoding temperature are comprehensive.

**Weaknesses:**

- The technical designs are rather trivial and technically unsound.
  - RAG with homologous proteins is a common practice in annotating protein databases [1]. I doubt if using only $k=3$ proteins with 30\% identity is sufficient for function annotation, as protein structure prediction models like AlphaFold2 [2] typically use tens to thousands of MSAs. Besides, it appears that the RAG module cannot handle orphan proteins [3] or poorly annotated homologs mentioned in Lines 42-43 either.
  - As shown in Algorithm 1 (Lines 7-33), since information from earlier attempts is not arranged into the context of further generation, the synthesis-and-judging workflow is simply an enumeration rather than a cascade. Moreover, the "early accept" operation compromises possible annotations with higher scores, and some of the models may rarely get used as the iteration of models is in the outer loop. I also doubt if the generations from the same model with a low temperature vary significantly.
  - In GO inference, it seems the GO hierarchy is simply used for information extraction and post-processing rather than guiding LLMs for generating more accurate GOs. Moreover, GO annotations in UniProtKB typically comprise only the leaf nodes, since their ancestors can be easily calculated with upper closure. This makes the formulation of Ontology consistency questionable, as predicting ancestors is redundant.
- The presentation of the methodology and experiments is unclear. See below:
  - In lines 86-87, what's the definition of "low-resource protein function gap-filling"?
  - What's the "unchar" set in Table 1 used for?
  - What's the meaning of - in Equation 2?
  - What's the data format of candidates $\hat{s}$ in Section 2.4? Are they simply texts or comprise other metadata?
  - What's the meaning of "allowlist" and "anchors" in Lines 216-217? How are they obtained?
  - What's the meaning of "per-aspect caps" in Line 248?
  - Where are the SCP results and K-sweep PR results mentioned in Lines 287-293?
  - The main document does not direct readers to the implementation details of baselines.
  - The settings of some critical hyperparameters like $L, T, \tau$ are not reported.
- The baselines and the adopted LLM variants are weak. For protein function description generation, multi-modal approaches like ProtLLM [4], ProteinGPT [5], and ProtChatGPT [6] should be compared. For GO prediction, deep learning models like DeepGO [7] and fine-tuned protein language models like ESM [8] should be compared. The strongest text model is GPT4o-mini, which "serves as an upper-bound reference", and the number of parameters of other LLMs does not exceed 8b. Considering the short context for each agent, I doubt if the cost of using API calls for stronger models like DeepSeek-V3.1 or GPT-5 is indeed unacceptable. The authors should at least provide comparisons of these models as their dataset is rather small, and justify that deploying open-sourced models with GPUs is more suitable than making API calls under "low-resource scenarios".

Refs.

[1] https://www.uniprot.org/help/automatic_annotation

[2] Highly accurate protein structure prediction with AlphaFold

[3] Structure prediction for orphan proteins

[4] ProtLLM: An Interleaved Protein-Language LLM with Protein-as-Word Pre-Training

[5] ProteinGPT: Multimodal LLM for Protein Property Prediction and Structure Understanding

[6] ProtChatGPT: Towards Understanding Proteins with Large Language Models

[7] DeepGO: predicting protein functions from sequence and interactions using a deep ontology-aware classifier

[8] Evolutionary-scale prediction of atomic-level protein structure with a language model

**Questions:**

My major concerns have been listed in the Weaknesses above. Here are some minor questions:
- Why do authors choose only 10 widely-studied species for dataset construction? Will this lower the difficulty of the task? Do model performances vary across different species?
- How frequently is the fallback agent called? How is the performance of this agent?
- What are the precision and recall scores in GO prediction? What's the error mode of the model (e.g., mostly false positives or true negatives)?
- Why do authors set 2% as the threshold for the consistency of GO prediction? Are there any biological insights behind this choice?
- What are the ablation results by removing different agents?
- The authors mentioned a trade-off between text quality and GO accuracy, which is counterintuitive, as a better function description may lead to more accurate GO annotations. Explanations are expected.

---

### Official Review · Reviewer_rFao · 2025-10-28

**Soundness:** 2
**Presentation:** 2
**Contribution:** 3
**Rating:** 4
**Confidence:** 4

**Summary:**

This paper introduces ProtFunAgent, an agentic LLM framework designed to improve protein function prediction, particularly in low-resource settings where homology transfer is unreliable and LLMs tend to hallucinate. The authors claim three key innovations of ProtFunAgent: (1) homology-guided RAG, which retrieves functional evidence from top-\(k\) sequence homologs; (2) ontology-constrained decoding, which aligns predictions with the Gene Ontology hierarchy via lexicon-aware filtering and pruning; and (3) a synthesis-judging cascade, where multiple LLMs collaborate and self-evaluate to refine candidate summaries. The system is evaluated on UniProt-derived benchmarks and is shown to outperform single-LLM and heuristic baselines, achieving over 3$\times\$ higher hierarchical F1 and nearly doubling recall while maintaining precision. The authors claim that ProtFunAgent closes more than half of the gap to oracle-level annotation, demonstrating that embedding biological structure into agentic LLM pipelines enables scalable, ontology-faithful function prediction.

**Strengths:**

1. The integration of agentic LLM cascades with structured biological priors (homology evidence, GO hierarchy) is a novel and promising direction for protein function prediction.

2. The paper includes extensive experiments comparing ProtFunAgent against multiple baselines (single-LLM variants, heuristic lower bounds, oracle upper bounds) and ablations (synthesis backbones, judge selection, temperature).

3. The work addresses a critical gap in automated protein annotation by combining generative flexibility with symbolic constraints, making it relevant for both computational biology and structured reasoning in LLMs.

**Weaknesses:**

1. While the paper compares against single-LLM variants and oracles, it does not include strong non-LLM baselines (e.g., DeepGOPlus, deepNF, or TAWFN) in the main results table. This makes it difficult to assess whether the gains are due to the agentic design or simply the use of LLMs.
2. The paper includes ablations on model backbones, judge selection, and temperature, but lacks a full component-wise ablation (e.g., removing RAG, removing ontology constraints, synthesis-judging cascade). The impact of the claimed three key innovations is not systematically ablated.
3. The use of multiple LLM calls (cascades, judges, fallbacks) may be computationally expensive. The paper does not discuss runtime, cost, or scalability, which are important for real-world deployment.
4. The paper lacks sections of Related Work and Conclusion.
5. Key hyperparameters (e.g., BLAST thresholds \(k=3\), \(\theta_{\text{id}}=30\%\), \(\theta_E=10^{-5}\), temperature settings) are provided, though more details on model-specific settings (e.g., judge thresholds) would be helpful.
6. The paper does not mention whether code will be released, which is critical for reproducibility given the complexity of the pipeline.

**Questions:**

1. How does ProtFunAgent compare to prior state-of-the-art non-LLM protein function predictors (e.g., DeepGOPlus, deepNF, TAWFN) in terms of GO prediction accuracy and ontology consistency?
2. Could you provide an ablation study that separately evaluates the contributions of the claimed three key innovations: Homology-RAG, ontology-constrained decoding, and the synthesis-judging cascade?
3. The paper mentions using a “compact lexical allowlist” in the constrained synthesis regime. How is this allowlist constructed, and how sensitive is the model to its composition?
4. What are the computational and latency costs of running the full ProtFunAgent pipeline, especially when using cascaded LLMs and multiple judge iterations?

---

### Official Review · Reviewer_nFdy · 2025-10-28

**Soundness:** 2
**Presentation:** 2
**Contribution:** 1
**Rating:** 2
**Confidence:** 3

**Summary:**

The manuscript proposes ProtFunAgent, a framework for protein function prediction. The key components include homology-guided retrieval-augmented generation, ontology-constrained decoding, and a synthesis-and-judging cascade of LLMs. The method demonstrates promising performance in protein function annotation tasks.

**Strengths:**

- The topic is important and relevant to the protein modeling community.
- Experimental results indicate that the proposed method achieves good performance on benchmark datasets.

**Weaknesses:**

- The work relies entirely on existing publicly available LLMs; there is no methodological novelty in model architecture or training.
- The contribution is mainly in system integration and application design, which lacks sufficient AI innovation for a top-tier AI conference.

**Questions:**

What is the main algorithmic contribution of the manuscript beyond applying existing LLMs with domain-specific heuristics?

---

### Official Review · Reviewer_UeEK · 2025-11-01

**Soundness:** 2
**Presentation:** 3
**Contribution:** 3
**Rating:** 4
**Confidence:** 4

**Summary:**

This paper presents ProtFunAgent, a agentic framework for protein function annotation. It integrates homology-guided retrieval to provide sequence context, synthesis and judging agents to generate and filter functional summaries, and ontology-constrained decoding to ensure Gene Ontology (GO) consistency. Evaluated on UniProt-derived benchmarks, the system shows improve performance than single-model or heuristic baselines while maintaining strong ontology adherence. The study introduces additional metrics such as ontology consistency and support-calibrated precision to better capture biological validity.

**Strengths:**

- Integration of biological prior and llm reasoning:
Combines homology retrieval, ontology constraints, and multi-stage agentic loop in a coherent framework for protein annotation.

- Clear pipeline design:
The method is organized into well-defined stages (synthesis, judging, ontology decoding) that are easy to understand and replicate conceptually.

- Promising performance gains:
Shows consistent improvements in GO prediction and recall compared to single-model and heuristic baselines.

- New evaluation metrics:
Introduces ontology consistency and support-calibrated precision to assess structural accuracy more comprehensively.

- Relevant application focus:
Targets low-resource and poorly annotated proteins, making the approach useful for practical biological curation.

**Weaknesses:**

- Data contamination risk: The benchmark is derived from UniProt, which overlaps with content likely included in LLM pretraining corpora. This makes it unclear whether ProtFunAgent is truly generalizing or simply recalling known annotations.

- Lack of evaluation on novel or post-cutoff proteins: The test proteins are clustered by 60% identity to prevent within-split leakage, but there is no evaluation on proteins released after the LLMs' training cutoff or from truly novel metagenomes. This limits confidence in out-of-distribution generalization.

- Lack of comparison against SoTA Models: Although DeepGOPlus, DeepGOZero, and TAWFN are cited as representative baselines, no quantitative results are reported. Without this, it is unclear how ProtFunAgent performs relative to the strongest baseline (supervised) predictors.

- Reliance on homology retrieval (SwissProt): Because the retrieval database is SwissProt (also the benchmark source), homology overlap may introduce indirect label leakage, i.e.,  retrieved homolog summaries could already include the ground-truth GO annotations for the query protein.

- Incomplete error analysis: While ontology consistency is measured, the paper lacks a breakdown of where errors occur, such as confusion between GO namespaces, hallucinated functions, or failures due to homolog scarcity.

- Missing resource and running time reporting: Without these, scalability and efficiency are hard to assess (moreover, the method relies on BLASTP for homology search, which is often an expensive process for large databases).

- Limited ablation on ontology-constrained decoding:  It would be important to isolate its contribution quantitatively since this component is central.

- Lack of downstream task validation: the generated GO sets are not tested in downstream pipelines (e.g. enrichment analysis, PPI prediction etc.).

**Questions:**

- How do the authors ensure that ProtFunAgent's results are not influenced by potential UniProt data seen during LLM pretraining?

- Can the authors provide runtime or computational cost measurements for this agentic pipeline?

- Can the authors provide quantitative comparisons with supervised baselines like DeepGOPlus or DeepGOZero to clarify performance differences?

- Have the authors considered validating the predicted functions through perturbation or motif-level analyses to confirm biological correctness?

---

### Note · Authors · 2025-12-05

I have read and agree with the venue's withdrawal policy on behalf of myself and my co-authors.